# Mpox Person-to-Person Transmission—Where Have We Got So Far? A Systematic Review

**DOI:** 10.3390/v15051074

**Published:** 2023-04-28

**Authors:** Pedro Pinto, Miguel Alves Costa, Micael F. M. Gonçalves, Acácio Gonçalves Rodrigues, Carmen Lisboa

**Affiliations:** 1Division of Microbiology, Department of Pathology, Faculty of Medicine, University of Porto, 4200-319 Porto, Portugal; pedrofilipegp@hotmail.com (P.P.); mfgoncalves@med.up.pt (M.F.M.G.); agr@med.up.pt (A.G.R.); 2Department of Dermatology and Venereology, Centro Hospitalar Vila Nova de Gaia/Espinho, 4434-502 Porto, Portugal; miguelalvescosta@gmail.com; 3Centre for Health Technology and Services Research/Rede de Investigação em Saúde (CINTESIS@RISE), Faculty of Medicine, University of Porto, 4200-319 Porto, Portugal; 4Department of Dermatology and Venereology, University Hospital Centre of São João, 4200-319 Porto, Portugal

**Keywords:** Mpox, Monkeypox, transmission, sexual, skin, surfaces, fomites, respiratory

## Abstract

The recent multi-country outbreak of Mpox (Monkeypox disease) constituted a public health emergency. Although animal-to-human transmission is known to be the primary way of transmission, an increasing number of cases transmitted by person-to-person contact have been reported. During the recent Mpox outbreak sexual or intimate contact has been considered the most important way of transmission. However, other routes of transmission must not be ignored. The knowledge of how the Monkeypox Virus (MPXV) spreads is crucial to implement adequate measures to contain the spread of the disease. Therefore, this systematic review aimed to collect scientific data published concerning other implicated sources of infection beyond sexual interaction, such as the involvement of respiratory particles, contact with contaminated surfaces and skin-to-skin contact. The current study was performed using the guidelines of Preferred Reporting Items for Systematic Reviews and Meta-Analyses (PRISMA). Publications analyzing contacts of Mpox index cases and their outcome after contact were included. A total of 7319 person-to-person contacts were surveyed and 273 of them became positive cases. Positive secondary transmission of MPXV was verified after contact with people cohabiting in the same household, with family members, with healthcare workers, or within healthcare facilities, and sexual contact or contact with contaminated surfaces. Using the same cup, sharing the same dishes, and sleeping in the same room or bed were also positively associated with transmission. Five studies showed no evidence of transmission despite contact with surfaces, skin-to-skin contact, or through airway particles within healthcare facilities where containment measures were taken. These records support the case for person-to-person transmission and suggest that other types of contact beyond sexual contact pose a significant risk of acquiring the infection. Further investigation is crucial to elucidate MPXV transmission dynamics, and to implement adequate measures to contain the spread of the infection.

## 1. Introduction

Mpox (Monkeypox disease) is a zoonotic infectious disease established in humans characterized by rash, fever, skin lesions, lymphadenopathy, headache, upper respiratory symptoms, oral ulcers, vomiting, and conjunctivitis [1]. This disease is caused by the Monkeypox virus (MPXV), a virus that belongs to the genus *Orthopoxvirus* and the family *Poxviridae* [2]. Although this virus was originally found in animals, it was firstly identified in humans in 1970 in the Democratic Republic of the Congo [3,4,5]. Historically, MPXV genetic diversity has been classified into two clades [6]. However, a new proposal for MPXV classification has been established into three clades (I, IIa, and IIb). Clade I corresponds to the prior “Congo Basin clade”; while clades IIa and IIb correspond to the “West Africa clade” [7]. The first outbreak of human Mpox outside its endemic region, registered in the USA, was indirectly related to various infected rodent species that were imported from West Africa. This outbreak occurred in prairie dogs living in contact with the rodent’s species imported from Ghana. As a result, the prairie dogs transmitted MPXV to approximately 40 humans. This was the first known outbreak of Mpox outside of Africa [8,9]. In November of 2022, the top 10 countries with reported Mpox cases were USA, Brazil, Spain, France, the United Kingdom, Germany, Colombia, Peru, Mexico, and Canada [10].

Primary animal-to-human infection is assumed to occur when handling Mpox infected animals through direct (touch, bites, or scratch) [11] or indirect contact, although the exact mechanism(s) remains to be defined. The virus is assumed to enter the body through broken skin, respiratory tract, or mucous membranes (eyes, nose, or mouth) [10]. Secondary human-to-human transmission occurs presumably through large respiratory droplets or direct or indirect contact with body fluids, lesion material, and contaminated surfaces or other material, such as clothing or linens [12]. Prolonged contact with patients renders hospital staff and family members at a greater risk of infection [13]. The risk of nosocomial transmission, vertical transmission, and fetal deaths have already been described [14,15]. One health approach is necessary for disease detection, including wildlife surveillance and investigation into animal reservoir.

This systematic review aims to summarize the evidence associated with the human-to-human transmission of the MPXV, and other sources of infection beyond sexual transmission, such as the involvement of respiratory particles, the contact with contaminated surfaces, and skin-to-skin contact.

## 2. Materials and Methods

### 2.1. Protocol

This systematic review was conducted based on the recommendations presented in the Preferred Reporting Items for Systematic Reviews and Meta-Analysis (PRISMA) guidelines [16]. The methodology used, including the screening of the title and abstract, full text reading, inclusion/exclusion, and data extraction criteria were accepted by all authors.

### 2.2. Eligibility Criteria

This systematic review was performed to be an updated review to collect scientific published data concerning other implicated sources of Mpox infection beyond sexual interaction. Articles meeting the following criteria were eligible for inclusion: (1) studies concerning people who contacted with cases of Mpox, and (2) whose outcome after contact was evaluated. We included publications without restrictions regarding the type of classification of the studies. Thus, articles referring to human-to-human contact, letters to the editors, observational studies, and other systematic reviews were included. Furthermore, there were no restrictions on the language or related to the date of publication of the articles. Publications up to the date of 15 April 2023were searched and included. Articles addressing animal-to-human or human-to-animal contact, vertical transmission, reviews, and case reports were excluded.

### 2.3. Information Sources and Search Strategy

We performed an electronic search on four databases, PubMed, Scopus, Web of Science, and Preprints (date of initial search 1 November 2022; last update 15 April 2023). To ensure that the search was inclusive, keywords (alone or in combination) related to scientific literature concerning Mpox person-to-person transmission were included. For this research, we used the MeSH Browser from the NIH (National Institute of Health) that allows a user to search directly for MeSH terms. The queries used are presented in Appendix A and included the terms: “monkeypox”, “transmission”, “transmissions”, “transmissibility”, transmissible”, “clothing”, “clothes”, “contact”, “surface”, “surfaces”, “fomite”, “fomites”, “particle”, “particles”, “sexual”, “mucous”, “contaminated”, “contamination”, “respiratory”, “skin”, “skin-to-skin”, “humans”, and “human”.

### 2.4. Study Selection

Two independent researchers (P.P. and M.A.C.) selected the articles one by one for inclusion or exclusion based on the title and abstract. The conflicts were solved by a third researcher (C.L.). The full text of the resulting articles was analyzed by two researchers, who selected independently the articles for inclusion. Once more, potential conflicts were discussed and solved by a third author.

An online program (Covidence systematic review software, Veritas Health Innovation, Melbourne, Australia) was used to remove duplicates, to conduct the screening of the articles, and to find the ones eligible for full text analysis.

### 2.5. Data Collection Process

Three authors (P.P., M.A.C. and M.F.M.G.) extracted information independently from the included articles and reached a consensus for data inclusion. Data extraction was conducted using Microsoft Word software and predefined tables. The following information was collected from each included study: author, year of publication, study design, setting data, case definition, type of contact/mode of transmission, index case(s), contacts (possible), secondary cases, conclusion, and limitations.

### 2.6. Study Quality Assessment

The risk of bias was evaluated using the quality assessment tool from the National Institutes of Health, which allowed us to classify the studies into poor, fair, or good quality depending on the fulfilment of the criteria defined by this tool [17]. This assessment was performed by M.A.C., M.F.M.G., and C.L. Divergences resulting from this process were discussed between the authors.

## 3. Results

### 3.1. Number of Retrieved Papers

A total of 2087 articles were found in our research on 15 April 2023, 740 of which were duplicated. Out of 1347 articles, 1233 were excluded after analysis of title and abstract and 114 were included for full-text analysis. We further excluded 99 articles considering inclusion and exclusion criteria. A final number of 15 observational studies were included in this systematic review: five retrospective cohorts and ten prospective cohorts. These results are summarized in Figure 1.

### 3.2. Study Characteristics

This systematic review included 15 studies [18,19,20,21,22,23,24,25,26,27,28,29,30,31,32] (Table 1): seven studies were from Africa [18,19,20,21,23,24,25], six of which from Zaire, now called Democratic Republic of the Congo [18,19,20,21,24], three studies from the USA [22,29,30], one from Singapore [26], and three studies from Europe, one from United Kingdom [27] and two from Spain [28,32], respectively. One study had no reference to the country of origin [31]. Ten studies were published before 2022 [18,19,20,21,22,23,24,25,26,27], and the remaining five were published in 2022 or 2023 [28,29,30,31,32].

### 3.3. Outcome Assessment and Evidence of Contact with MPXV

The clade of MPXV involved was only known in one study, which was the West African clade [29]. In two studies, the outcome assessment was only clinical [26,27]. Another study does not specify how the outcome was assessed [31]. The other 12 had, at some point, confirmation by serological findings or virus isolation [18,19,20,21,22,23,24,25,28,29,30]. All studies reported person-to-person contact with infected patients. A total of 7319 contacts were surveyed, and 273 of them were considered Mpox positive cases. Seven studies reported household contact [18,19,20,21,23,24,28], six studies reported surface/fomites contact [22,26,28,30,31,32], two of them at a tattoo studio [28,32], two studies reported potential contact with airway particles [22,30] and three reported skin-to-skin contact [22,30,31], two studies reported intrafamilial contact [23,25], one study reported sexual contact [28], one study reported direct contact with the patient or the patient’s surroundings or specimens [26], and one study reported person-to-person but no sexual or skin-to-skin contact [29]. Two studies focused on Mpox transmission on pediatric population [23,28].

### 3.4. Evidence of Transmission

Ten studies showed positive secondary transmission of MPXV [18,19,20,21,23,24,25,27,28,32]. In these, there was evidence of contact with people cohabiting in the same household [18,19,20,21,23,24,28], with family members [23,25], with healthcare workers [25,27], or within healthcare facilities [23], with contaminated surfaces at a tattoo studio [28,32] and sexual [28]. Drinking from the same cup, eating from the same dish, sleeping in the same room, and sleeping in the same bed were positively associated with transmission [24]. Five studies showed no evidence of transmission [22,26,29,30,31] despite contact with surfaces, skin-to-skin contact, or through airway particles in five healthcare centers [22,26,27,30,31] and one in a prison facility [29].

Data related to the use of mask, gloves, or other protective equipment were available in 7 of the 15 studies [22,23,26,27,29,30,31], 2 of which reported low usage of protective measures [23,29].

### 3.5. Risk of Bias within and across Studies

A total of six studies were considered to have poor quality [18,19,23,25,28,32], seven studies were considered fair [22,24,26,27,29,30,31], and two studies were considered as good [20,21]. The studies with poor quality consequently resulted in low levels of evidence. Participants were lost to follow-up in some studies and others had a very low population size. The same cases were probably present in different included studies, which could result in duplicate information. The use of gloves and other protective equipment made it somewhat difficult to measure adequately the exposure of contacts. A previous smallpox vaccination history could underestimate the risk of transmission to unvaccinated individuals. Outcomes were not always obtained by PCR techniques, having been evaluated only symptomatically, which meant that outcome measures were not consistently applied within and across studies. Additionally, the studies are not entirely comparable because they occurred in different places, at different moments in time, and they must be analyzed taking these questions into account.

## 4. Discussion

### 4.1. Summary of Evidence

Although several studies suggested a household transmission of the virus, the type of contact occurring in the habitation is difficult to understand [18,19,20,21,23,28]. The transmission of MPXV could be a result of skin-to-skin contact, respiratory particles, sharing the same household surfaces or fomites, or intimate contact. For instance, Nolen et al. [24] showed that people who share the same cup and dish, and sleep in the same room or bed are prone to transmitting the virus to others. Such outcome corroborates the assumption that MPXV might be transmitted by surfaces or fomites. Nevertheless, activities such as kissing or laundering clothes showed no correlation with the acquisition of the virus [24]. Although the type of transmission is not clear, these studies reinforce the possibility of person-to-person transmission of MPXV. However, it was reported that people who shared the same space with a confirmed Mpox case in a prison facility in Chicago—despite some of them being offered post-exposure prophylaxis—did not reveal positive transmission after contact with the index patient [29]. None of the exposed reported sexual contact with others nor skin-to-skin contact; they only reported washing their clothes in communal showers, sharing personal hygiene items, or sitting on other’s beds. Therefore, additional data on the type of contact between people who live in the same household are needed.

Some studies showed no evidence of transmission of the virus after exposure of healthcare workers to infected patients through potentially contaminated surfaces and fomites, airway particles like during aerosol-generating procedures, skin-to-skin contact [22,26,30,31], and through handling patient’s specimens [26]. The use of protective measures like masks, gloves, and medical gowns by healthcare workers while caring for patients with Mpox can protect themselves from infection [22,26,30,31]. Nevertheless, transmission to healthcare workers was documented by Besombes et al. [25], in Central African Republic and to other patients in a hospital in Democratic Republic of Congo by Learned et al. [23]. We highlight the importance of educating people and healthcare personnel to prevent infectious outbreaks by strengthening health centers’ capacity and resources in remote forest areas as well as to learn measures to control the transmission to themselves and others.

Besombes et al. [25] and Learned et al. [23] showed also a positive intrafamilial transmission and a cascade of spread over three and six waves of person-to-person transmission, respectively, which shows the implication of other types of transmission other than sexual contact: from an index patient to his two daughters, two sisters, and one sister-in-law [25] and from at least two contacts in a family that ended up having Mpox confirmed [23]. Person-to-person transmission was also verified in the pediatric population [28], aged 7 months to 17 years old, which was related to possible sexual transmission, transmission via contaminated material in a tattoo and piercing studio, or by contact with their parents. There is evidence that MPXV can be widely spread through contaminated materials at tattoo and piercing studios [28,32], which is a reminder that surfaces must be cleaned after usage at these places.

After the beginning of the recent Mpox outbreak in 2022, a study conducted by Vaughan et al. [33] assessed the current epidemiological situation, based on confirmed cases of Mpox in 36 countries in the European region submitted to the European Surveillance System (TESSy). The authors demonstrated that the majority of MPXV cases were likely to be transmitted sexually. Other cases were described to involve a surface/fomite, non-sexual, and non-healthcare related transmission. However, the contact with clear confirmed cases was not possible in this study given that the analyses performed were based on data submitted to TESSy database.

Regarding the transmission through surfaces, four studies [34,35,36,37] evaluated the presence of MPXV on surfaces in the household [34,35], workplace [36], and a hospital environment [37] after a positive case was in close contact with those surfaces. All studies showed high amounts of detectable virus on surfaces directly touched by a person with Mpox, and the virus could be present either on porous or nonporous surfaces [34,35]. When the specimens were collected by swabbing the surfaces were cultivated on cell cultures (Vero cells), the virus was showed to be viable in some specimens in two studies [34,37] and not viable in any specimen in one study [35]. Morgan et al. [34] stated that the virus viability can be maintained for at least a period of 15 days after the contact. Therefore, these results show that MPXV could infect people through contact with surfaces. Although this hypothesis has not been confirmed yet, Mpox-specific cleaning, maintaining appropriate hand hygiene, and decontamination measures of surfaces should be considered in such situations [35,36]. Nevertheless, additional studies regarding MPXV transmission via contaminated surfaces and objects are crucial since there is still insufficient knowledge on this topic.

Due to a lack of data regarding the transmission of MPXV, it is not clear what has changed concerning the transmission dynamics comparing Mpox cases before and after the outbreak of 2022. Although it has been shown that individuals were more prone to acquire infection following sexual contact, we need more clear evidence that clearly states that.

A recent study by Al-Raeei [38] looked at the contagiousness of MPXV and has given a basic reproduction number related to its transmissibility considering the recent world outbreak. They found that the average R0 number, after evaluating the dynamics of one country of each continent was 1.2810. As R0 is >1, it means that the epidemic is evolving rather than plateauing. The estimated number of R0 is important to know at what stage of outbreak we are in and what can be done to contain it.

### 4.2. Limitations

This systematic review poses several limitations. Some studies showed low quality or did not mention what type of person-to-person contact played a role in the transmission [18,19,23,25,28]. Moreover, other studies which contained only epidemiological data and lacked information on the type of contact between two infected individuals were excluded, which may bias the results. Additionally, the 2022 outbreak is a recent occurrence and thus, there is still little information, which does not address the knowledge gaps about Mpox transmission dynamics.

There is also an additional risk of bias in establishing the transmission potential of the virus because some patients received post-exposure prophylaxis [29], while others were vaccinated against MPXV [18,19,20,21,22,25,27,29,30]; in other studies, information about previous vaccination was lacking.

Some of the included articles did not confirm a Mpox infection among contacts by detection of MPXV DNA using the PCR technique. That can lead to a bias of the results because it is possible that some infected people might be asymptomatic. Some individuals, especially healthcare workers, used protection like masks and gloves while making contact with a positive case. That probably limited the spread of infection, and the results could be biased [21,26,27,29,30]. People might have made contact in different ways at the same time, thus making it difficult to know which type of contact was responsible for the infection.

## 5. Conclusions

The recent outbreak of MPXV still is of great concern. Cases of infection have been reported in several countries, worldwide, and the virus transmission initially started as person-to-person contact. This systematic review aimed to clarify the different types of transmission that have been established. The findings of this systematic review support the view that MPXV can be transmitted person-to-person in addition to sexual contact, by piercing and tattooing, contaminated surfaces, objects, and fomites.

Nevertheless, considering the limited studies on Mpox in humans, there is a need for an improvement in the quality of studies and further investigation focusing on understanding the types of transmission of the virus. It is very important to know how individuals can protect themselves from infection and which precautions are needed. Yet, the use of masks is recommended when in contact with any suspect or confirmed case, especially in healthcare facilities. Activities such as frequent hand disinfection and appropriate surface cleaning are widely recommended.

In future investigations, it is necessary to maintain a good follow-up of case contacts and to correctly establish the type of contact between people. In addition, confirmation of an Mpox infection by PCR should be considered in all studies.

## Figures and Tables

**Figure 1 viruses-15-01074-f001:**
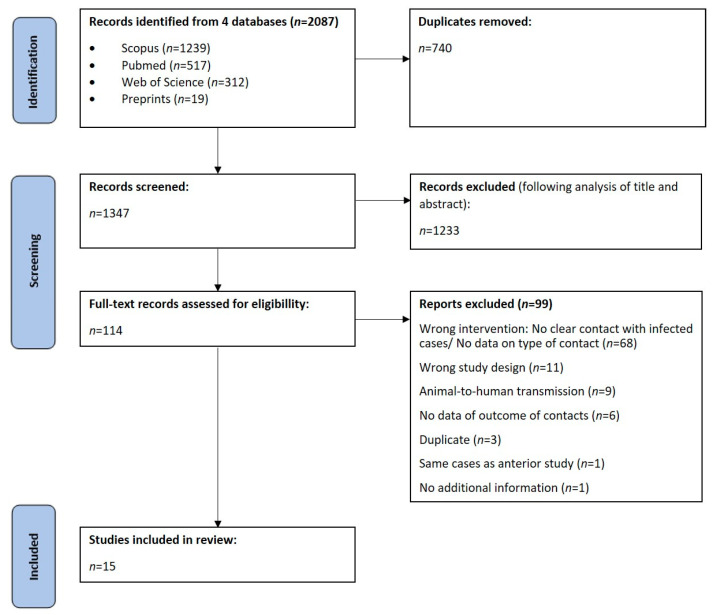
Flow chart of study selection.

**Table 1 viruses-15-01074-t001:** Characteristics of the 13 included studies.

Authors, Year	Type of Study	Setting Data	Case Definition	Type of Contact/Mode of Transmission	Index Cases	Possible Contacts	Secondary Cases	Conclusions	Limitations	Quality Assessment ^1^
Arita et al. [18]	Retrospective cohort (observational)	West and Central Africa1970–1983 Data included are from Zaire	Clinical, epidemiological and laboratorial diagnosis	Person to person	--	Household: 708 Vaccination scar: 566 No vaccination scar: 142 Other: 945 Vaccination scar: 710 No vaccination scar: 235	Household: 25 Vaccination scar: 3 No vaccination scar: 22 Other: 6 Vaccination scar: 0 No vaccination scar: 6	The attack rate for household contacts was significantly higher than that for other contacts.	No data on how surveillance of contacts and cases was carried out. No case definition.	Poor
Jezek et al. [19]	Retrospective cohort (observational)	Outbreak of 5 children of human MPXV belonging to 2 families West Kasai, Zaire May-July 1983	Clinical diagnosis + Serological test: 1 case Virus isolation from skin lesion: 3 cases	Person to person	1 index case (animal to human) initiated a cascade of transmission that ended up with 4 secondary or tertiary infected cases.	Household: 17 Vaccination scar: 6 No vaccination scar: 11 Other: 22 Vaccination scar: 2 No vaccination scar: 20	Household: 3 Vaccination scar: 2 No vaccination scar: 1 Other: 1 Vaccination scar: 1 No vaccination scar: 0	Epidemiologic investigations of the 5 children suggest that the 1st case was infected from an animal source, and the other 4 cases were infected from a previous human case.	No case definition. Possibly, the same cases are reported by Arita et al. [18].	Poor
Jezek et al. [20]	Prospective cohort study	Evaluation of transmission to 2510 contacts of 214 patients Zaire 1980–1984	Confirmed case was one in which the clinical and epidemiological diagnosis of human MPXV was confirmed by laboratory findings	Person to person	214 cases primary cases: 152 secondary cases: 62	Household: 1187 Vaccination scar: 910 No vaccination scar: 277 Other: 1323 Vaccination scar: 959 No vaccination scar: 364	Household: 48 Vaccination scar: 14 No vaccination scar: 34 Other: 14 Vaccination scar: 2 No vaccination scar: 12	The attack rate for household contacts was significantly higher than that for other contacts, among both unvaccinated (4 times higher) and vaccinated (7 times higher) household contacts.	Only a few of the vaccinated contacts were tested serologically to determine a subclinical infection. Possibly, the same cases are reported by Arita et al. [18].	Good
Jezek et al. [21]	Prospective cohort study	Evaluation of transmission from 338 patients Zaire 1981–1986	A patient having onset of rash between 1 and 3 weeks after exposure to an index case is a secondary case, which may have arisen by person-to-person transmission. Confirmation by laboratory findings	Person to person	338 cases primary cases: 245 secondary cases: 93	Household: 1420 Vaccination scar: 989 No vaccination scar: 431 Other: 858 Vaccination scar: 566 No vaccination scar: 292	Household: 53 Vaccination scar: 13 No vaccination scar: 40 Other: 16 Vaccination scar: 2 No vaccination scar: 14	The affected households appear to be the most important focal point for dissemination of MPXV to susceptible individuals.	Possibly, the same cases are reported by Jezek et al. [20]	Good
Fleischauer et al. [22]	Prospective cohort study: analysis of exposure of HCWs to hospitalized infected patients	Evaluation of MPXV infection in HCWs who were within a radius of 2 m from 3 infected patients USA/Indiana 2003	Serological studies were carried out. Presence of fever or rash with at least 2 other symptoms, including chills, headache, backache, lymphadenopathy, sore throat, cough, and shortness of breath	Person to person	Hospitalized patients: 3	HCWs contacts: 57 Vaccination: 31 No vaccination: 26 Gloves: 35 Gown: 19 Surgical mask: 14 N95 mask: 11 Surfaces/ fomites: 46 Airway particles: 52 Skin-to-skin: 28 Out of 57 HCWs, 40 (70%) had ≥1 unprotected exposure	Symptoms: 0 Both IgG and IgM positive: 0 Just IgM: 1, was vaccinated in the last 6 months. 1 positive IgM, (asymptomatic)	Transmission not verified in healthcare facilities.	Potential selection bias (the participation rate in hospitals A and B was different). A control group was not used to test the accuracy of the ELISA test—there is the potential of some false positive tests. Potential reporter bias—difficulty to recall.	Fair
Learned et al. [23]	Retrospective Cohort	Cascade of six sequential transmissions in a hospital and between people belonging to the same family, almost all were less than 18 years old. Democratic Republic of the Congo 2003	Suspect and probable case—meets both epidemiological and symptomatic (fever or rash if suspect, both conditions if probable) criteria Confirmed case—meets epidemiological, symptomatic and laboratory (virus in culture, PCR or immunohistochemical) criteria	Person to person	Index case: 1 Initiated a cascade of transmission that led to 11 more secondary cases: 1 suspect 7 probable 3 confirmed Type of contact: Household visitor: 1 (also owned a pet monkey, in good health) Hospital contact with a case: 3 Intrafamilial contact with a case: 2 Both intrafamilial and hospital contact: 4 Not specified: 1	--	Surveillance was not undertaken among exposed HCWs. However, none acquired symptoms.	Identified 11 clinical cases of secondary MPXV infection spread over 6 waves of person-to-person transmission. Transmission verified in healthcare facilities between patients, where control measures were not present. The primary infected people were children (less than 18 years old) After control measures were taken, no more cases were identified. No access to protective equipment (gloves, masks…).	No data on vaccination. Healthcare workers must have been spared because of previous vaccination.	Poor
Nolen et al. [24]	Retrospective cohort study	Identification of specific activities and behaviors potentially associated with an increased risk of MPXV transmission within 16 household Democratic Republic of the Congo 2013	Clinical diagnosis or PCR (Cases that were not confirmed by PCR were based only on symptoms)	Person to person	Primary cases 17	Household total: 97	MPXV symptomatic contacts: 44 (22% vaccinated) Positive correlation: Drinking from the same cup Eating from the same dish Sleeping in the same room Sleeping in the same bed Negative correlation: Laundering clothes Kissing Assisting with toileting and hygiene	Risk factors of acquiring MPXV in a household included sleeping in the same room or bed or using the same plate or cup as the primary case. Activities associated with an increased risk of MPXV transmission all have potential for virus exposure to the mucosa.	Only 30% of the cases were confirmed by PCR. Possibility of immunity because of previous infection with MPXV.	Fair
Besombes et al. [25]	Prospective cohort study (observational) analysis of exposure of contacts to an index case	Intrafamily transmission on a family from an index case Central African Republic 2018	Clinical diagnosis + laboratory confirmation (PCR)	Person to person	Index case (animal to human transmission): 1 Intrafamilial contact (confirmed by PCR): 5 Case-patient’s daughters: 2 Case-patient’s sisters: 2 Case-patient’s sister-in-law: 1	HCWs contacts: 2 Other case patient’s family: 5 Case-patient’s village contacts: 31	Total: 6 cases positive by serological test (all asymptomatic) HCWs: 2 (1 vaccinated) Other case patient’s family: 3 (1 animal contact and vaccinated) (1 vaccinated) (1 no animal contact, not vaccinated) Case-patient’s village contacts: 1	Identified 5 clinical cases of secondary MPXV infection spread over 3 waves of intrafamilial infection, originating from an index case patient with primary infection possibly attributable to contact with wild fauna. Positive serological findings in healthcare workers highlight the limited infection prevention and control resources, to protect HCWs responding to outbreaks in Central African Republic.	No data about the kind of intrafamily transmission. No data about HCWs’ protection.	Poor
Kyaw et al. [26]	Prospective cohort study (observational) analysis of previous contacts of index cases	HCWs who were in contact with a case of MPXV, before admission to the isolation unit Singapore 2019	Clinical diagnosis	Person to person	Hospitalized patient: 1 traveler who had recently returned from Nigeria (probable animal to human transmission)	Close contact: 27 Direct contact with the patient himself or the patient’s surrounding: 12 Surfaces/fomites (handled the patient’s linen and cleaned the NEP room): 3 Laboratory staff who had handled the patient’s specimens: 12 All had protected exposure to the patient, with the appropriate and adequate use of PPE.	All asymptomatic	Clear infection prevention guidelines on the appropriate PPE for different HCWs, based on patient care activities and the transmission risk are crucial.	HCWs used protection like gloves and face masks while having contact with the patient, so results must be biased. No lab confirmation that 27 HCWs had MPXV infection. No sampling of objects or surroundings of the index case patient.	Fair
Vaughan et al. [27]	Prospective cohort study (observational) analysis of contacts of 1 HCW	HCW who contact with a case of MPXV infected from Nigeria United Kingdom 2018	Clinical diagnosis	Person to person Direct exposure of skin lesions, body fluids, including clothing or bedding without wearing appropriate PPE	Hospitalized patient: 1 traveler who had recently returned from Nigeria	134 Adequate protective measures were taken (gown, gloves, face shield)	4 became ill	The use of standard PPE may not have afforded sufficient protection against MPXV particularly if skin lesion debris containing the virus had been disturbed and inhaled when bedsheets were changed. The risk to the public was very low because effective human-to- human transmission requires close contact with an infected person or virus-contaminated materials.	No serology tests. No data on previous smallpox vaccination. Post-exposure vaccination to MPXV could have inhibited disease.	Fair
Aguilera-Alonso et al. [28]	Prospective cohort study	Pediatric population (n = 16) Aged 7 months to 17 years old Spain, 2022	Clinical diagnosis + laboratory confirmation (PCR)	Sexual	Total: 16 positive cases 3 (aged 13–17 years old)	--	--	Either sexual, contact with contaminated material in a tattoo studio, or household contact with parents caused infection. Sexual contact does not explain all infections by MPXV.	Data were collected by epidemiological surveys based on interviews with patients and their family members.	Poor
				Surfaces/ fomites (contaminated material in a tattoo and piercing studio)	9 (aged 13–17 years old)	--				
				Household (contact with their parents: 3)	3 (age inferior 4 years old)	--				
				Unknown	1 (age inferior 4 years old)	--				
Hagan et al. [29]	Prospective cohort study	Contacts with a confirmed case in a jail Clade: West African USA/Chicago 2022	Clinical diagnosis, serological test, or both	Person to person but no sexual/skin-to-skin contact	1 resident	57 residents (22 lost follow-up) 35 residents completed follow-up (Mask usage is probably low)	Symptoms + serological testing for only 14 residents that consented to testing IgM positive: 0 IgG positive: 3 (asymptomatic)	No evidence of skin-to- skin or sexual contact among residents. No secondary cases were identified.	Difficult quantification of exposure risk. Out of 57 potentially exposed, only 35 completed follow-up. Out of 36 potentially exposed, only 13 accepted post-exposure prophylaxis; only 14 accepted testing. 3 IgG positive probably because of previous vaccination. Serological testing performed 7 days after potential exposure for some residents, when they might not yet have seroconverted. Self-report of symptoms/sexual contact. Not able to confirm childhood smallpox vaccination history.	Fair
Marshall et al. [30]	Prospective cohort study	HCWs exposed to 55 patients with MPXV USA/Colorado 1 May–31 July 2022	Clinical diagnosis + laboratory confirmation (PCR) in two people	Surfaces/fomites	Total: 55 patients	Total: 313 HCWS 12% received postexposure vaccine. N = 26 HCWs Glove use: 23 No glove use: 3 Unknown glove use: 0	Presence of symptoms: 7 2 people who had rash or lesions (n = 3) performed a PCR test: all negative	No HCWs developed an MPVX infection during the 21 days after exposure. Infection prevention training is important in all healthcare settings, and these findings can guide future updates for PPE recommendations and risk classification in healthcare settings.	Selection bias—these results must not be generalized to the community population. Data related to previous vaccination are lacking. Lack of information about exposure to contaminated materials. Lack of information about the use of facemasks by patients.	Fair
				Airway particles (Aerosol-generating procedures)		N = 7 HCWs Mask use during the procedure: 3				
				Skin to skin		N = 161 HCWs Glove use: 125 No glove use: 30 Unknown glove use: 6				
Phelippeau et al. [31]	Prospective cohort study	HCWs exposed to a patient with MPXV	Unknown	Person to person	1 patient	26 HCWs had direct contact Skin: 5 Clinical examination: 2 Undressing/making bed/temperature/blood pressure: 1 Linen: 4 Talking: 1 Eye care: 7 Measuring blood pressure/temperature: 3 Transport: 1 Other: 1 Glove use: 10 No glove use or other: 16 Vaccinated: 9	None	No secondary cases were identified. At this hospital, HCWs were at low risk of contracting the infection.	It is not known if the assessment of outcomes was clinical or serological. Post-exposure vaccination to MPXV could have inhibited disease.	Fair
Martinez et al. [32]	Retrospective cohort study	Infection in customers at a tattoo parlor Spain 6 July–19 July 2022	PCR	Person to person Surfaces (contaminated material in a tattoo and piercing studio)	Unknown	58 customers	21 cases 20 after piercing 1 after tattooing Parlor staff: no cases	21 secondary cases were identified after contact with infected material at the tattoo parlor. The mode of transmission was probably the direct contact after piercing and tattooing. Material used at tattoo places must be carefully disinfected.	Unknown index case. No data on parlor staff protection (masks). No serological testing to evaluate asymptomatic MPXV infection.	Poor

--: Not mentioned; MPXV: Monkeypox virus; HCWs: Healthcare workers; ^1^: Quality Assessment Tool for Observational Cohort and Cross-Sectional Studies.

## Data Availability

Not applicable.

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
