# Peer review of "Mpox Person-to-Person Transmission—Where Have We Got So Far? A Systematic Review"

_viruses, 2023, doi:10.3390/v15051074_

Round 1

Reviewer 1 Report

The analysis of literature data given in this article is of some interest, but the actual material is poor, which reduces the quality of this article.

Text notes: 1.      Lines 42-43. Instead of ‘Monkyepox Virus’, you should write ‘Monkeypox virus’. 2.     Line 45. References should be added to articles describing the discovery of MPXV in human samples for the first time. 3.     Lines 48-49. In 2003, the first cases of monkeypox in humans outside the African continent were reported in the United States.

      4. Line 57. References should be added.

Author Response

The analysis of literature data given in this article is of some interest, but the actual material is poor, which reduces the quality of this article.

Reply: Thank you for the comments, the effort and the time taken to review the manuscript. All suggestions were carefully considered.

Text notes:

  1. Lines 42-43. Instead of ‘Monkyepox Virus’, you should write ‘Monkeypox virus’.

Reply: Revised as suggested.

  1. Line 45. References should be added to articles describing the discovery of MPXV in human samples for the first time.

Reply: Revised as suggested.

  1. Lines 48-49. In 2003, the first cases of monkeypox in humans outside the African continent were reported in the United States.

Reply: The sentence has been clarified.

  1. Line 57. References should be added.

Reply: The sentence has been rewritten.

Reviewer 2 Report

The manuscript “Mpox person-to-person transmission – Where have we got so far? A systematic review” aimed to collect scientific data published concerning other implicated sources of infection beyond sexual interaction. The authors conclude that other types of contact beyond sexual contact compose significant risk to acquire the infection, and further investigation is crucial to elucidate MPXV transmission dynamics and to implement adequate measures to contain the spread of the infection.

Mpox is primarily endemic to remote regions of the rainforest of West and Central Africa, where it has been mainly transmitted from the animal reservoir(s) to humans. Since May 2022, this viral disease of zoonotic origin represents the most recent PHEIC, when thousands of cases of human Mpox have been identified in several non-endemic countries, marked by human-to-human transmission, and caused by people’s movement at a global level, suggesting changes both in biological aspects of the virus and in human behaviour. So, as with all zoonotic diseases, a comprehensive One Health approach is necessary for disease detection, including wildlife surveillance and investigations in animal reservoir. This will need to keep in mind in order to address the knowledge gaps about Mpox transmission recent outbreak of MPXV stating “The findings of this systematic review support that MPXV can be transmitted person-to-person besides sexual contact, by piercing and tattooing, contaminated surfaces, objects, and fomites” as Public Health Threat forgetting the Neglected Tropical Disease. Several outbreaks are ongoing in some countries of West and Central Africa driven by a combination of population growth, accumulation of unvaccinated smallpox population, and a decline in smallpox vaccine immunity. These outbreaks have resulted from a complex intersection of events and, given the zoonotic nature of the disease, required a robust outbreak response collaboration among humans, animals, and environmental health experts.

The introduction should provide a clear statement of the problem, and so some suggestions are bellow presented:

Lines 46-48 – Historically, MPXV genetic diversity has been classified into two clades. However, a new proposal for MPXV classification has been established into three clades (I, IIa, and IIb). Clade I correspond to the prior “Congo Basin clade”; while clades IIa and IIb correspond to the “West Africa clade”.

Lines 48-50 – First outbreak of human monkeypox outside its endemic region, registered in the USA was indirectly related to various rodent species infected imported form West Africa. This outbreak has occurred in prairie dogs living in contact with the rodent’s species imported from Ghana. As a result, the prairie dogs transmitted MPXV to approximately 40 humans. This was the first known outbreak of Mpox outside of Africa.

Lines 50-53 – As on September of 2022 the top 10 countries with reported monkeypox cases were USA, Spain, Brazil, France, Germany, United Kingdom, Peru, Colombia, Mexico, and Canada.

Lines 54-66 – Primary animal-to-animal infection is assumed to occur when handling monkeypox infected animals through direct (touch, bites, or scratch) or indirect contact, although the exact mechanism(s) remains to be defined. The virus is assumed to enter the body through broken skin, respiratory tract, or mucous membranes (eyes, nose, or mouth). Secondary human-to-human transmission occurs presumably through large respiratory droplets or direct or indirect contact with body fluids, lesion material, and contaminated surfaces or other material, such as clothing or linens. Prolonged contact with patients renders hospital staff and family members at a greater risk of infection. The risk of nosocomial transmission, vertical transmission, and fetal deaths have already been described.

Lines 66-68 – One Health approach is necessary for disease detection, including wildlife surveillance and investigation into animal reservoir.

Line 187 – What does it mean “A previous vaccination history could underestimate the risk of transmission to unvaccinated individuals”. May be related with the smallpox vaccination scar, as described in columns Possible contacts and Secondary cases of Table 1.

Following minor corrections

Line 118 – Please change to “…in our first research on the three databases,…”

Line 123 – Please change to “…in this systematic review: four retrospective cohorts and nine prospective cohorts.”

Lines 145-149 – Please change to “…(Table 1): seven studies were from Africa [10-13, 15-17], six of which from Zaire, now called Democratic Republic of the Congo [10-13, 16], three studies from the USA [14, 21, 22], one from Singapore [18], and two studies from Europe, respectively from United Kingdom and Spain [19, and 20]. Ten studies were published before 2022 [10-19], and the remaining three were published in 2022 [20-22].

Lines 157 and 158 – Please change to “…was only known in one study, which was the West African clade [21]. In two studies, the outcomes…”.

Lines 162-166 – Please change to “…four studies reported surface/fomites contact [14, 18, 20, and 22], two studies reported potential…, two studies reported intrafamilial contact [15, and 17], one study reported sexual contact [20], one study reported direct contact…, one study reported person-to-person…”

Line 176 – Please change to “…through airway particles in three healthcare centers [14, 18, and 21…”

Line 178 – Please change to “…in six of the 12 studies [14, 15, 18, 19, 21, and 22], two of which…”

Lines 181 and 182 – Please change to “A total of five studies were considered… to have poor quality [10, 11, 15, 17, and 20], six studies were considered fair [14, 16, 18, 19, 21, and 22], and two studies were good [12, and 13].

Line 224 – Please change to “…and a cascade of spread over three and six waves of…”

Lines 226 and 227 – Please change to “…an index patient to his two daughters, two sisters and one sister-law [17], and from at least two contacts in a family…”

Line 228 – Please change to “…aged seven months to 17 years old…”

Line 239 – Please change to “Regarding the transmission through surfaces, four studies [24-27] evaluated…”

Author Response

The manuscript “Mpox person-to-person transmission – Where have we got so far? A systematic review” aimed to collect scientific data published concerning other implicated sources of infection beyond sexual interaction. The authors conclude that other types of contact beyond sexual contact compose significant risk to acquire the infection, and further investigation is crucial to elucidate MPXV transmission dynamics and to implement adequate measures to contain the spread of the infection.

Mpox is primarily endemic to remote regions of the rainforest of West and Central Africa, where it has been mainly transmitted from the animal reservoir(s) to humans. Since May 2022, this viral disease of zoonotic origin represents the most recent PHEIC, when thousands of cases of human Mpox have been identified in several non-endemic countries, marked by human-to-human transmission, and caused by people’s movement at a global level, suggesting changes both in biological aspects of the virus and in human behaviour. So, as with all zoonotic diseases, a comprehensive One Health approach is necessary for disease detection, including wildlife surveillance and investigations in animal reservoir. This will need to keep in mind in order to address the knowledge gaps about Mpox transmission recent outbreak of MPXV stating “The findings of this systematic review support that MPXV can be transmitted person-to-person besides sexual contact, by piercing and tattooing, contaminated surfaces, objects, and fomites” as Public Health Threat forgetting the Neglected Tropical Disease. Several outbreaks are ongoing in some countries of West and Central Africa driven by a combination of population growth, accumulation of unvaccinated smallpox population, and a decline in smallpox vaccine immunity. These outbreaks have resulted from a complex intersection of events and, given the zoonotic nature of the disease, required a robust outbreak response collaboration among humans, animals, and environmental health experts.

Reply: Thank you for the comments, the effort and the time taken to review the manuscript. We are grateful for the insightful comments. All suggestions were carefully considered.

The introduction should provide a clear statement of the problem, and so some suggestions are bellow presented:

Lines 46-48 – Historically, MPXV genetic diversity has been classified into two clades. However, a new proposal for MPXV classification has been established into three clades (I, IIa, and IIb). Clade I correspond to the prior “Congo Basin clade”; while clades IIa and IIb correspond to the “West Africa clade”.

Reply: Thanks! We added in the introduction.

Lines 48-50 – First outbreak of human monkeypox outside its endemic region, registered in the USA was indirectly related to various rodent species infected imported form West Africa. This outbreak has occurred in prairie dogs living in contact with the rodent’s species imported from Ghana. As a result, the prairie dogs transmitted MPXV to approximately 40 humans. This was the first known outbreak of Mpox outside of Africa.

Reply: We added in the introduction.

Lines 50-53 – As on September of 2022 the top 10 countries with reported monkeypox cases were USA, Spain, Brazil, France, Germany, United Kingdom, Peru, Colombia, Mexico, and Canada.

Reply: We added in the introduction. We changed September to November according to WHO (the top 10 countries were the same but in a different order).

Lines 54-66 – Primary animal-to-animal infection is assumed to occur when handling monkeypox infected animals through direct (touch, bites, or scratch) or indirect contact, although the exact mechanism(s) remains to be defined. The virus is assumed to enter the body through broken skin, respiratory tract, or mucous membranes (eyes, nose, or mouth). Secondary human-to-human transmission occurs presumably through large respiratory droplets or direct or indirect contact with body fluids, lesion material, and contaminated surfaces or other material, such as clothing or linens. Prolonged contact with patients renders hospital staff and family members at a greater risk of infection. The risk of nosocomial transmission, vertical transmission, and fetal deaths have already been described.

Reply: We added in the introduction.

Lines 66-68 – One Health approach is necessary for disease detection, including wildlife surveillance and investigation into animal reservoir.

Reply: We added in the introduction.

Line 187 – What does it mean “A previous vaccination history could underestimate the risk of transmission to unvaccinated individuals”. May be related with the smallpox vaccination scar, as described in columns Possible contacts and Secondary cases of Table 1.

Reply: The sentence has been clarified. People who contacted in some way with people infected with Monkeypox virus must not have acquired the infection, not because the virus is not transmissible, but because people are protected against it with previous smallpox vaccination.

Following minor corrections

Line 118 – Please change to “…in our first research on the three databases,…”

Reply: Revised as suggested.

Line 123 – Please change to “…in this systematic review: four retrospective cohorts and nine prospective cohorts.”

Reply: Revised as suggested.

Lines 145-149 – Please change to “…(Table 1): seven studies were from Africa [10-13, 15-17], six of which from Zaire, now called Democratic Republic of the Congo [10-13, 16], three studies from the USA [14, 21, 22], one from Singapore [18], and two studies from Europe, respectively from United Kingdom and Spain [19, and 20]. Ten studies were published before 2022 [10-19], and the remaining three were published in 2022 [20-22].

Reply: Revised as suggested.

Lines 157 and 158 – Please change to “…was only known in one study, which was the West African clade [21]. In two studies, the outcomes…”.

Reply: Revised as suggested.

Lines 162-166 – Please change to “…four studies reported surface/fomites contact [14, 18, 20, and 22], two studies reported potential…, two studies reported intrafamilial contact [15, and 17], one study reported sexual contact [20], one study reported direct contact…, one study reported person-to-person…”

Reply: Revised as suggested.

Line 176 – Please change to “…through airway particles in three healthcare centers [14, 18, and 21…”

Reply: Revised as suggested.

Line 178 – Please change to “…in six of the 12 studies [14, 15, 18, 19, 21, and 22], two of which…”

Reply: Revised as suggested.

Lines 181 and 182 – Please change to “A total of five studies were considered… to have poor quality [10, 11, 15, 17, and 20], six studies were considered fair [14, 16, 18, 19, 21, and 22], and two studies were good [12, and 13].

Reply: Revised as suggested.

Line 224 – Please change to “…and a cascade of spread over three and six waves of…”

Reply: Revised as suggested.

Lines 226 and 227 – Please change to “…an index patient to his two daughters, two sisters and one sister-law [17], and from at least two contacts in a family…”

Reply: Revised as suggested.

Line 228 – Please change to “…aged seven months to 17 years old…”

Reply: Revised as suggested.

Line 239 – Please change to “Regarding the transmission through surfaces, four studies [24-27] evaluated…”

Reply: Revised as suggested.

Reviewer 3 Report

Thank you for allowing me to review your paper. I appreciate the interesting data presented, but I believe there are opportunities for improvement to make your analysis more robust.

Overall, the results resemble those of a bibliometric study rather than a systematic review. To make the analysis more comprehensive, it is necessary to include more diverse and internationally recognized databases. The databases you used index practically the same content. Additionally, it would be helpful to include preprints and organize the paper more effectively as there are several sentences that do not make sense or are out of context.

There are also fundamental errors in English and study selection.

Specifically, the method used for the systematic review is inconsistent. What reference did you use to conduct your systematic review? The date of November 2022 is too outdated for a subject that is constantly evolving. The date should be at least February 2023.

The search terms used in the databases vary and are not explicit. It is not clear what types of studies should be included, as well as the inclusion and exclusion criteria. More clarity is needed in this regard.

I hope these comments are helpful in improving your paper

Author Response

Thank you for allowing me to review your paper. I appreciate the interesting data presented, but I believe there are opportunities for improvement to make your analysis more robust.

Overall, the results resemble those of a bibliometric study rather than a systematic review. To make the analysis more comprehensive, it is necessary to include more diverse and internationally recognized databases. The databases you used index practically the same content. Additionally, it would be helpful to include preprints and organize the paper more effectively as there are several sentences that do not make sense or are out of context.

There are also fundamental errors in English and study selection.

Specifically, the method used for the systematic review is inconsistent. What reference did you use to conduct your systematic review? The date of November 2022 is too outdated for a subject that is constantly evolving. The date should be at least February 2023.

The search terms used in the databases vary and are not explicit. It is not clear what types of studies should be included, as well as the inclusion and exclusion criteria. More clarity is needed in this regard.

I hope these comments are helpful in improving your paper

Reply: Thank you for the comments, the effort and the time taken to review the manuscript. Thank you for pointing out the strengths of this study. All Materials and Methods section was thoroughly and carefully revised as suggested. We conducted this systematic review based on the recommendations presented on Preferred Reporting Items for Systematic Reviews and Meta‐Analysis (PRISMA) guidelines. Considering that our review aimed to search for other implicated sources of Mpox infection beyond sexual interaction, we excluded other studies whose outcomes could not be of interest for this work, for being self-reported or not using objective measures.

We used the keywords (alone or in combination) related to scientific literature concerning Mpox person-to-person transmission. For this research, we used the MeSH Browser from the NIH (National Institute of Health) that allows to search directly for MeSH terms. We added this information in the methodology as suggested.

We updated the search again using the 3 electronic databases and we included relevant studies published between 1 November 2022 and 15 April 2023.

We also search in preprints database for articles that meet the inclusion criteria that have been outlined, but none were added.

The inclusion and exclusion criteria section were improved as suggested.

Round 2

Reviewer 2 Report

The manuscript was improved by the careful review done by the authors. Now, the manuscript is suitable to be published, after one minor correction:

Line 90 - Please correct to "...an electronic search on four database..."